# An exploratory study on midwives' perceptions of the availability of preconception care in health facilities in the Kpandai District of Northern Ghana

Richard Kwaku Bawah[1]☯*, Gilbert Abotisem Abiiro[2‡], Gifty Apiung Aninanya[3‡], Barnabas Sackeyuah[4]☯

1 Division of Nursing & Midwifery, Department of Pediatrics, Evangelical Church of Ghana Hospital, Kpandai, Ghana, 2 Department of Population and Reproductive Health, School of Public Health, University for Development Studies, Tamale, Ghana, 3 Department of Global and International Health, School of Public Health, University for Development Studies, Tamale, Ghana, 4 Division of Public Health, Department of Health Information, Evangelical Church of Ghana Hospital, Kpandai, Ghana

☯ The authors contributed equally to this work.
‡ GAA and GAA also contributed equally to this work.
* rkbawah@gmail.com

## Abstract

### Background

Preconception care is a critical component of reproductive health that significantly influences the well-being of individuals and couples prior to conception. Midwives play a critical role in addressing potential health problems prior to pregnancy, with the priority of optimizing positive pregnancy outcomes. However, midwives in developing nations, including Ghana, often face significant challenges in delivering preconception care services.

### Aim

This study explored midwives' perceptions of the availability of preconception care in health facilities in the Kpandai District of Northern Ghana.

### Method

An exploratory qualitative research design was used in conducting the study. Both purposive and random sampling techniques were used in recruiting 15 midwives from 25 health facilities. Data saturation was achieved on the 14th participant; hence, no further interviews were conducted. Interviews were audio-recorded and transcribed verbatim. Thematic content analysis was employed to analyze the data.

**Data availability statement:** All relevant data are within the paper and its Supporting Information files.

**Funding:** The authors received no specific funding for this work.

**Competing interests:** The authors have declared that no competing interests exist.

## Results

The participants exhibited adequate knowledge, positive attitudes, and appropriate practices regarding the delivery of preconception care. However, they reported a low level of client uptake of these services. Notably, the majority of participants were unaware of any standardized guidelines or protocols governing preconception care. Despite these challenges, participants generally expressed satisfaction and a strong willingness to provide preconception care services within their respective units.

## Conclusion

The study underscores the necessity for the development and implementation of clear guidelines and standardized protocols to ensure consistent and effective delivery of preconception care. Such measures are essential to guarantee that all individuals of reproductive age, especially those with pre-existing health conditions, receive appropriate preconception care interventions prior to conception.

## Introduction

Preconception care (PCC) improves pregnancy outcomes and reduces maternal and neonatal mortalities [1]. The World Health Organization (WHO) defines PCC as interventions optimizing women's and couples' health before conception, covering biomedical, nutritional, behavioral, and social aspects [2]. M'hamdi and colleagues, Poels and collegues [3,4] emphasize PCC as a primary prevention strategy designed to educate individuals of reproductive age and manage modifiable health risks. Intervening during the preconception period allows for the identification and optimization of health conditions, thereby improving the chances of favorable pregnancy outcomes [5,6].

The Sustainable Development Goals (SDGs) 3 and 5 emphasize the attainment of universal access to sexual and reproductive health services and rights by 2030, with particular focus on reducing maternal and neonatal mortality [7]. These objectives are closely aligned with the UN Secretary-General's Global Strategy for Women's, Children's, and Adolescents' Health [8]. The revised Global Strategy sets ambitious targets, including ensuring that no country records a maternal mortality ratio (MMR) greater than 140 per 100,000 live births or a neonatal mortality rate (MMR) exceeding 12 per 1,000 live births. Currently, there are stake disparities in MMR worldwide, ranging from as low as 3 per 100,000 live births in countries like Finland, Greece, Iceland, and Poland, to as high as 1,360 per 100,000 live births in countries such as Finland, Greece, Iceland, and Poland, to as high as 1,3360 per 100,000 live births in Sierra Leone [9]. In Ghana, a total of 860 maternal deaths were reported in 2023, with 100 occurring in the Northern Region [10]. Neonatal mortality trends exhibit similar inequities, with rates as low as 3.0 per 1,000 live births in high-income countries, compared to 23.2 per 1,000 in South Asia and 25.9 per 1,000 in sub-Saharan Africa [11]. Ghana recorded 4,038 neonatal deaths in 2023, including 342 in the Northern

Region and 17 in the Kpandai District [10]. These persistently high maternal and neonatal mortality rates are largely attributed to inadequate attention to healthcare prior to conception [12,13].

Healthcare providers (HCPs) play an essential role in the delivery of preconception care (PCC) through a range of services, including outpatient consultations, outreach initiatives, and clinical care across all stages of life, from adolescence through prenatal, delivery, and postnatal periods [14–16]. The effectiveness of PCC implementation is closely tied to the level of knowledge HCPs possess regarding its concept and components. Evidence suggests that knowledge among HCPs about PCC remains suboptimal. For instance, a study conducted in Ethiopia reported that only 31% of healthcare providers demonstrated adequate knowledge of PCC [17]. Similarly, colleagues [18] identified a significant lack of awareness among midwives regarding preconception health. In another study, only 52% of healthcare providers were found to have good knowledge of PCC [19]. A qualitative study in Ghana further highlighted that many providers lacked the necessary knowledge and competence to deliver effective PCC services [20]. In addition to knowledge, the attitudes of midwives and other HCPs toward PCC are crucial determinants of service quality. Positive attitudes can significantly influence the uptake and delivery of PCC services [21]. Midwives' perspectives and approaches play a pivotal role in shaping care delivery and improving outcomes for individuals of reproductive age [4]. Research consistently shows that nurses and midwives generally hold favorable attitudes toward PCC delivery [22–24]. For example, a study in the Netherlands found that midwives exhibited moderately positive attitudes toward preconception consultations, although certain gaps remained [24]. Similarly, another study reported that midwives recognized the value of PCC in enhancing maternal and child health outcomes and generally expressed supportive attitudes toward its implementation [25]. In Ethiopia, 71% of obstetric care providers were found to have a positive attitude toward PCC [21], underscoring the potential for improving service delivery through targeted capacity-building and educational initiatives

Despite generally positive attitudes toward preconception care (PCC), many healthcare providers face significant challenges in delivering high-quality PCC services. Comparative analyses across various countries reveal substantial disparities in the provision of PCC by healthcare professionals. For instance, in Nevada, only 43% of providers reported offering PCC services, whereas in Ontario, Canada, approximately 96% of family physicians provided such care. In Iran, the proportion of providers delivering PCC ranged between 25% and 47.6%, while in the Netherlands, only 27% of general practitioners and 20% of midwives engaged in PCC delivery. In Egypt and Ethiopia, PCC provision was reported at approximately 15% and 15.3%, respectively [26]. Similarly, a study focusing on obstetric healthcare providers found that only 34.5% demonstrated adequate practice in PCC [21]. A qualitative study in the Netherlands also revealed that a small proportion of primary caregivers actively delivered PCC services [24]. Multiple factors contribute to this suboptimal practice of FCC, despite positive attitudes. Goossens and colleagues [27] identified that some healthcare providers exhibited unfavorable attitudes toward PCC, often attributed to limited knowledge and ambiguity regarding their responsibilities in delivering such care. Additionally, a lack of familiarity with the components of PCC remains a major barrier to effective service delivery [18,20,28].

In the Kpandai District of Northern Ghana, there is a notable absence of qualitative studies exploring the perceived determinants of PCC delivery. Existing research has predominantly focused on antenatal care clients and has highlighted midwives' limited access to PCC guidelines [29,30]. M'hamdi and colleagues [4] further emphasized that the absence of a structured PCC program, clear guidelines, and institutional support significantly hinders implementation among healthcare providers. Moreover, conflicting perspectives among providers regarding reproductive autonomy, pregnancy planning, and role responsibility in delivering PCC have compounded the challenges [31,32]. Given that the quality and consistency of PCC delivery are closely linked to healthcare providers' knowledge and attitudes, it is imperative to understand how these factors influence service provision. This study, therefore, sought to explore midwives' perceptions of the availability and delivery of preconception care services within healthcare facilities in the Kpandai District of Northern Ghana.

## Methods/Materials

### Study design

An exploratory qualitative research design was employed in this study, as it is well-suited for gaining in-depth insights into the personal experiences, perceptions, and contextual understanding of the study participants.

### The study setting

The study was conducted in selected health facilities in the Kpandai District of the Northern region of Ghana. The Kpandai District was selected as the study setting due to pressing reproductive and child health challenges, including teenage pregnancy, sexually transmitted infections, low first-trimester registration of pregnant women, poor uptake of family planning services, and unskilled deliveries. These issues may be linked to the limited provision and utilization of PCC services in the district. Preliminary data from the district health directorate reveal a lack of established guidelines or locally developed protocols for PCC delivery, hindering healthcare providers' ability to offer PCC services effectively, hence, the selection of the district as the study setting.

### Study population

The study focused on professional healthcare staff, particularly midwives, responsible for delivering maternal and child health services in the Kpandai District.

### Inclusion criteria

Midwives with at least one year of working experience in the maternal and child healthcare units and departments of health facilities in the Kpandai District were eligible for inclusion.

### Exclusion criteria

Registered nurses and auxiliary health staff, whether currently employed or not, in maternity, labor, antenatal care, and reproductive and child health clinics were excluded.

### Sampling techniques and procedures

Purposive sampling was used to select health facilities with midwives, while a combination of simple random and purposive sampling methods was employed to recruit midwives from these facilities within the district [33]. The district has 25 health facilities, all offering maternal and child health services. However, only those with registered midwives were purposively included in the study. Facilities were chosen based on their geographical distribution across communities to ensure fair representation. The description of the facilities included CHPS: Community-based Health Planning and Services, ECGH – Evangelical Church of Ghana Hospital, and KDH – Kpandai District Hospital.

According to data from the Kpandai District Health Directorate, there were 50 registered midwives working in the district. The number of midwives recruited for the study was determined by the proportionate distribution of midwives in each health facility. To calculate the number of participants, the number of midwives in each facility was divided by the total number of midwives in the district. In facilities with more midwives than required, a simple random sampling method was used, where midwives drew ballots marked "yes" or "no." Those who selected "yes" were included, while those who chose "no" were excluded. In facilities with only one midwife, that midwife was purposively selected, provided they consented to participate.

Purposive sampling was employed to select health facilities with certified midwives to ensure that the study focused on participants with direct experience and knowledge relevant to maternal and child health services. Given that the research aims to explore the perspectives and practices of midwives within the district, it was essential to target facilities providing

these services. The selection of facilities based on their geographical distribution ensured that data were gathered from diverse community settings, enhancing the representativeness of the study. Additionally, purposive sampling ensured that all selected facilities were staffed by qualified midwives, which is crucial for obtaining accurate and reliable data on PCC. This technique allowed the study to focus on key individuals who possess the requisite knowledge and experience, thereby contributing to the richness of the qualitative data collected.

Simple random sampling was used within facilities with more midwives than required to ensure fairness and reduce selection bias. By employing a lottery system (ballot drawing), each midwife had an equal chance of being selected, which enhances the credibility and generalizability of the findings within the district. This method aligns with the study's objective to capture a broad range of midwives' experiences and practices, thereby contributing to the breadth of the data collected. In facilities with only one midwife, purposive sampling ensured their inclusion due to their unique position and experiences in providing maternal and child health services. This integration of sampling methods allowed the study to include voices from both larger facilities with multiple midwives and smaller, single-midwife facilities, ensuring comprehensive coverage of the research phenomena.

The integration of purposive and random sampling techniques in this study provided a balanced approach to participant selection [33]. Purposive sampling ensured that only relevant facilities and qualified midwives were included, thereby enhancing the depth and relevance of the data. Random sampling, on the other hand, minimized bias and ensured that the study captured a wide range of experiences from midwives within larger facilities. This complementary approach facilitated the achievement of a comprehensive understanding of PCC practices in the district. It allowed the study to gather in-depth qualitative insights from purposively selected participants while also ensuring that the data were not skewed by selection bias, due to the random sampling method employed within larger facilities. Together, these methods provided a well-rounded and robust dataset, essential for addressing the study's objectives related to midwives' practices of PCC services. Further details on the sampling process are provided in Fig 1.

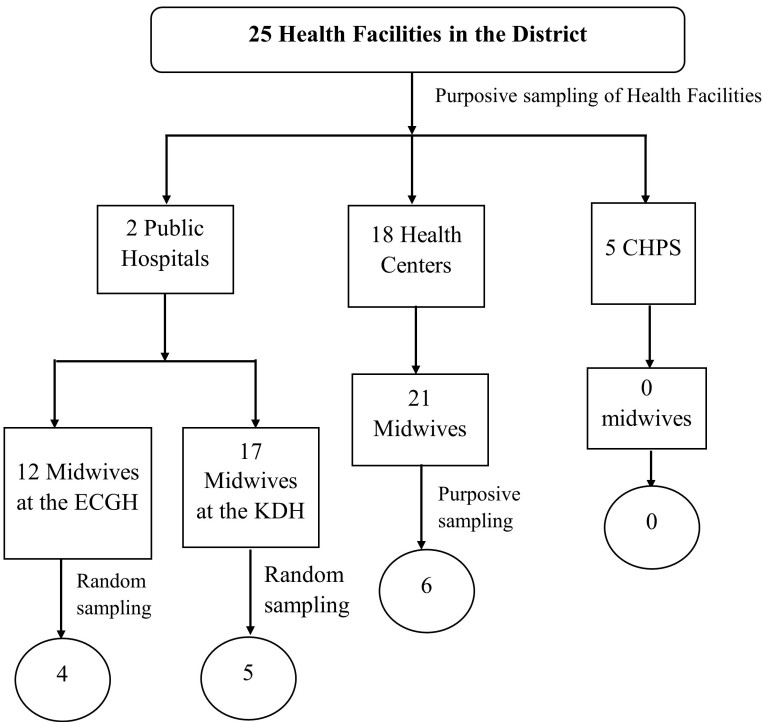

**Fig 1. Multi-stage sampling of study participants.**

## Data collection tool and procedures

An unstructured interview guide was used to collect data from the study participants through face-to-face interviews. The guide consisted of open-ended questions developed based on the study's objectives and literature review. It was divided into four sections: A, B, C, and D. Section A focused on participants' demographic information. Section B explored midwives' knowledge levels regarding PCC. Section C examined the attitudes of midwives towards PCC delivery. Section D investigated midwives' practices of PCC.

The contact details of the midwives in the district were obtained from the Kpandai District Health Directorate. Eligible participants were individually contacted at their respective health facilities and provided with a study information sheet and consent form. The information sheet outlined the study's purpose, potential risks, benefits, compensation, and participants' rights. Any questions or concerns raised by participants were addressed. Participants who signed the consent form were invited to designated offices in each health facility for a face-to-face interview to ensure privacy and confidentiality. All interviews were audio-recorded with participants' consent and conducted in English, with each lasting approximately 25 minutes. Fourteen participants were interviewed from the study setting between March 1 and May 31, 2024.

## Methodological rigor

The researchers adhered to various aspects of methodological rigor to enhance the credibility and validity of the study's outcomes. To ensure the reliability of the data collection instrument, a pre-test interview was conducted with 5 participants across 3 facilities. Transferability was addressed by providing detailed descriptions of the study's context, participants, and the systematic procedures used to generate the findings and conclusions. The research process was meticulously documented, including field notes, interview transcripts, and coding procedures, to ensure the dependability of the results. Additionally, the researchers performed thorough reviews of the study procedures, findings, and narratives to eliminate biases and maintain objectivity.

## Data analysis

Data analysis was conducted concurrently with data collection. The interview sections were audio-recorded and later transcribed verbatim. Each transcript was coded to ensure participants' anonymity. The coded transcripts were organized and analyzed using thematic analysis, as outlined by Braun and colleagues [34]. The analysis began with **Familiarization with data:** All audio-recorded interviews were transcribed verbatim on a Microsoft Word document, ensuring that each participant's identity remained anonymous [35,36]. RKB and BS repeatedly read the transcripts to become familiar with the data and gain a deep understanding of the content. **Generating Initial Codes:** Each transcript was manually coded line by line independently by RKB and BS using text boxes. Initial text codes were assigned to segments of data that were relevant to the research questions. Codes were short labels that captured the essence of specific pieces of text [37]. **Searching for Themes:** After coding all transcripts, similar text codes were grouped together to form potential themes. This step involved looking for patterns and relationships among the text codes [34,37]. **Reviewing Themes:** The potential themes were reviewed by comparing them against the entire dataset to ensure they accurately represented the data. Some themes were merged, refined, or discarded based on their relevance and frequency by both reviewers [37]. **Developing Major themes and Sub-themes:** Each theme was clearly defined and named (Major themes) to reflect the core idea behind them. Further repetitive words or phrases (Sub-themes) were also identified to capture more specific aspects within broader themes (Major themes) [38]. **Producing the Report:** A detailed report was produced, presenting the final themes with supporting quotes from the data. The report provided narratives that linked the themes to the study's objectives [36,37]. The research team, including GAA and GAA, collaboratively reviewed and validated the themes, narratives, and supporting quotes to ensure accuracy and coherence in the analyzed data.

## Ethical consideration

Ethical approval (CHRPE/AP/072/24) was granted by the Committee on Human Research and Publication Ethics of Kwame Nkrumah University of Science and Technology before data collection began. Formal permission was also obtained from the Kpandai District Health Directorate and the heads of the selected facilities. All participants provided informed consent and were aware of their right to withdraw at any time without legal consequences. The participants' anonymity and confidentiality were strictly maintained throughout the study.

## Results

### Demographics

A total of fifteen midwives were initially invited to participate in the study; however, data saturation was reached after the fourteenth interview, and thus the final participant was not included. The participants' ages ranged from 28 to 37 years. The duration of professional practice varied, with the shortest being 1 year at the staff midwife level and the longest being 8 years at the senior midwifery officer level. Of the fourteen participants, five were stationed in antenatal care units, four were assigned to maternity units at the hospital level, and the remaining five were based at health centers. Table 1 presents details of the sociodemographic characteristics of the study participants. ***Footnotes:*** P – Participant, SM – Staff Midwife, SSM – Senior Staff Midwife, MO – Midwifery Officer, SMO – Senior Midwifery Officer.

### Themes and sub-themes

Following the final coding and thematic analysis, three major themes and ten associated sub-themes emerged from the data. The first major theme, which addressed midwives' knowledge of preconception care, encompassed five sub-themes. The second theme, focusing on midwives' attitudes toward PCC, revealed three sub-themes. The third and final theme, which examined the actual delivery of PCC services among midwives, comprised two sub-themes. These themes collectively provide a comprehensive understanding of midwives' perspectives and practices related to PCC. This is shown in Table 2.

**Table 1. Participants sociodemographic data.**

| Codes | Age (years) | Gender | Marital status | Religion | Highest edu. level | Years of practice | Current rank | Unit/ department |
|---|---|---|---|---|---|---|---|---|
| P1 | 42 | F | Married | Christian | Degree in Midwifery | 8 | SMO | ANC |
| P2 | 28 | F | Married | Christian | Diploma in Midwifery | 5 | SSM | ANC |
| P3 | 34 | F | Single | Muslim | Degree in Midwifery | 1 | MO | Maternity |
| P4 | 29 | F | Married | Muslim | Diploma in Midwifery | 7 | SSM | Maternity |
| P5 | 33 | F | Married | Christian | Diploma in Midwifery | 4 | SSM | ANC |
| P6 | 26 | F | Single | Muslim | Diploma in Midwifery | 2 | SM | ANC |
| P7 | 34 | F | Married | Christian | Diploma in Midwifery | 4 | SM | Health Center |
| P8 | 37 | F | Married | Christian | Degree in Midwifery | 6 | MO | Maternity |
| P9 | 28 | F | Single | Christian | Diploma in Midwifery | 2 | SM | Health Center |
| P10 | 34 | F | Married | Muslim | Diploma in Midwifery | 5 | SSM | Maternity |
| P11 | 29 | F | Single | Muslim | Diploma in Midwifery | 1 | SM | Health Center |
| P12 | 37 | F | Married | Christian | Diploma in Midwifery | 4 | SSM | ANC |
| P13 | 32 | F | Single | Christian | Diploma in midwifery | 3 | SSM | Health Center |
| P14 | 29 | F | Single | Christian | Diploma in midwifery | 4 | SSM | Health Center |

**Table 2. Themes and sub-themes.**

| Main themes | Sub-themes |
|---|---|
| Midwives' knowledge regarding PCC | 1. Definitions of PCC<br>2. Components of PCC<br>3. Benefits of PCC<br>4. Category of individuals requiring PCC<br>5. Availability of standardized guidelines/protocol |
| Midwives' attitude regarding PCC | 1. Reception<br>2. Willingness<br>3. Communication |
| PCC delivery practices among midwives | 1. Provisions of PCC services<br>2. A dedicated unit for PCC |

Source: (Field Data, 2024).

**Midwives' knowledge regarding preconception care**

**Definition of PCC.** The majority of participants described PCC as a form of healthcare provided to individuals, regardless of marital status, who express the intention to conceive. They characterized PCC as a comprehensive approach that encompasses physical, psychological, and physiological aspects of health, aimed at optimizing the well-being of prospective parents prior to conception. The following participants made these comments regarding the definition of PCC:

*"… Is a care or counseling given to couples that are preparing to become pregnant…so basically you give them guidelines on how to care for their pregnancy till delivery."* [A Senior Staff Midwife at the ANC, P2]

*"…. Is a care that we give to couples or any other individuals… Even if you are not married and you are ready like to get pregnant. We counsel them on certain stuff so that they will prepare themselves and make the right decision towards pregnancy."* [A Staff Midwife at a Health Center, P9]

*"…. Is just like you preparing your client, let's say a client psychologically, mentally, and physically before pregnancy. It deals with you counseling the client, making her know her status…"* [A Staff Midwife at a Health Center, P11]

Two participants defined PCC services to include a time frame within which PCC services could be provided thus within 1–3 years before pregnancy:

*"… Is a care given to a woman before she conceives…. normally takes 1-3 years before the woman will conceive. She has to be on iron drugs or folic acid for not less than 3 months before she can conceive..."* [A Midwifery Officer at the Maternity, P3]

*"… Usually is a care that is rendered to women who intend to conceive not for now but probably the person has an intention of conceiving a year or 2 or more or when the person is married…."* [A Staff Midwife at the ANC, P6]

**Components of PCC.** Participants demonstrated varied levels of understanding regarding the components of preconception care (PCC). However, the majority identified several key elements as integral to PCC. These included health education, laboratory screening for both pre-existing and potential medical and obstetric conditions, blood grouping and assessment of rhesus factor incompatibility, family planning services, genetic counseling, psychological support, as well as nutritional assessment and counseling. This reflects a relatively broad conceptualization of PCC among the

midwives, encompassing biomedical, psychological, and preventive health components. The following participants stated the components of PCC:

> *"We screen the couple to make sure that they are compatible as in maybe blood grouping, then we also screen them like STIs, then we counsel them on the need to maybe prepare the woman for breastfeeding…. you have to prepare her mind and all that and then maybe, medications that she will be taking during the pregnancy."* [A Senior Staff Midwife at the ANC, P2]

> *"…. You have to know if there is any medical condition like diabetes, hypertension, sickle cell, HIV, then you know what to do."* [A Midwifery Officer at the Maternity, P3]

> *"You have to screen; you take their history as well. The history you can talk of social, occupation, then maybe if the woman is having any conditions, not the woman per say, but both couples. The history may be any surgical history or any medical condition which will inform us know how to manage the woman…."* [A Senior Staff Midwife at the Maternity, P4]

> *"So basically, we educate them on hygiene, nutrition then the kinds of medications to take before conceiving. So, we educate them on some reproductive hygiene."* [A Midwife Officer at the Maternity, P8]

> *"…. We do nutritional counseling, genetics, psychological counseling…"* [A Staff Midwife at a Health Center, P9]

One participant highlighted a component of PCC applicable between pregnancies, known as inter-conception care. The participant stated:

> *"We counsel them on family planning. That is maybe after delivery, they will have to accept family planning services so that they can space their children, so that will help them to take care of their children well."* [A Senior Staff Midwife at the ANC, P2]

**Category of people requiring PCC.** Most participants identified a broad range of individuals as eligible for PCC, highlighting its relevance beyond conventional reproductive groups. They emphasized that PCC should be provided to couples, both male and female, as well as to individuals who are single. Additionally, they recognized the importance of offering PCC to adolescents, adults, and individuals with comorbidities, reflecting an inclusive understanding of the target populations who may benefit from preconception interventions. The participants stated:

> *"… Every female who is up to the stage of reproductive age, everybody is entitled to especially like this supplement even at the adolescent age you can even start with the supplements, that is why we go to the schools especially the JSS and SSS to give them the IFA….as it prepares them towards conception."* [A Senior Midwifery Officer at the ANC, P1]

> *"… women who are not fertile and seriously searching for pregnancy…."* [A Senior Staff Midwife at the Maternity, P4]

> *"…and then elderly primid, after 35 years that is the time, the person is struggling to get her first pregnancy…."* [A Senior Staff Midwife at the Maternity, P10]

Some participants emphasized the need to prioritize specific populations for PCC due to their increased risk of adverse reproductive outcomes. They identified adolescents and individuals with comorbidities, such as HIV, sickle cell disease, diabetes mellitus, hypertension, hepatitis B, and various nutritional deficiencies, as requiring special attention. These groups were seen as needing targeted interventions to optimize their health status prior to conception and reduce potential complications during pregnancy. The following participants made these remarks:

*"Those who have HIV or diabetes need special attention, because when the baby is born the baby is supposed to be on medication, so it is important to give them special attention before they get pregnant, during pregnancy, and after they give birth."* [A Senior Midwifery Officer at the ANC, P1]

*"…. especially the adolescents, you know they are still schooling and they need to finish their education before they start maybe conception or giving birth. So, with those people, you emphasize regular care so that they don't involve themselves in any unwanted pregnancy which might affect their education."* [A Senior Staff Midwife at the ANC, P5]

*"…. especially those with HIV, when the person gives birth, a lot goes into it and the person would even need counseling before the person even gets pregnant because of the stigma and everything. The counseling would come in and then even during the pregnancy the medication the person would have to take and then right after birth, the medications the child will also be taking…."* [A Staff Midwife at a Health Center, P7]

**Benefits of PCC.** The participants exhibited a strong understanding of the benefits associated with PCC. Most participants identified key advantages, including the prevention of Rh incompatibility-related complications, reduction in the risk of preterm or premature labor, and the early detection and management of existing or potential health conditions in prospective mothers or individuals prior to conception. These insights reflect an appreciation of PCC's role in optimizing maternal and neonatal outcomes through proactive health interventions. These participants said:

*"…. some of the clients are there who do not go through preconception care…. when they finally realize that they are not compatible, then it becomes a problem because if we are able to detect any sickness or maybe disease early, we will be able to counsel them…. before maybe they marry…."* [A Senior Staff Midwife at the ANC, P2]

*"So, when you do those things, it can help to prevent these abnormalities and then also help the safe delivery of the mother, and then I may say, it can prevent preterm labor or premature labor."* [A Senior Staff Midwife at the Maternity, P4]

*"If we were to be taking preconception care more seriously here in Ghana, I think a lot of infertility problems would have been solved or prevented because the person would start early. There are so many STIs we don't know, we don't even know about. Before the person would know, it would have already caused havoc to the person's uterus….one of the benefits of preconception care is curbing infertility, reducing infertility…"* [A Staff Midwife at a Health Center, P7]

Two participants highlighted additional benefits of PCC, emphasizing its role in preparing individuals, particularly adolescents, to pursue their educational aspirations by promoting informed reproductive choices. They also noted that PCC equips adults for the physical, emotional, and psychological demands of pregnancy, childbirth, and the postpartum period, thereby enhancing their readiness for a healthy maternal life. The participants said:

*"… and the adolescents too, like I said, it will also enable them to finish their school before they will start giving birth."* [A Senior Staff Midwife at the ANC, P5]

*"….it can also reduce some congenital malformations such as the neural tube defects, which are caused by folic acid deficiency, so if the person starts taking the folic acid before getting pregnant, some of these congenital malformations wouldn't even be found, yeah."* [A Staff Midwife at a Health Center, P7]

**Availability of guidelines or protocols.** Nearly all participants reported a lack of awareness regarding the existence of standardized guidelines or protocols for preconception care (PCC) from key health authorities, including the World Health Organization (WHO), the Ministry of Health, the Ghana Health Service (GHS), or the Christian Health Association

of Ghana (CHAG). Despite this gap, participants indicated that they continued to provide PCC services based on the foundational knowledge acquired during their formal training in midwifery education. The participants said:

*"As for now, I've not…there is no protocol or guideline on that."* [A Senior Staff Midwife at the ANC, P2]

*"…. like we don't have a standardized protocol or guideline because we don't normally practice it."* [A Midwife Officer at the Maternity, P8]

*"There is no protocol, but from school knowledge, that's what we use to provide the preconception care."* [A Senior Staff Midwife at the ANC, P12]

*"I've not seen any preconception guidelines. What I do is that, what I've learnt and the little knowledge I have is what I depend on to provide the care. Since I started working, they've never provided any preconception guidelines to us."* [A Senior Staff Midwife at a Health Center, P14]

**Midwives' attitude regarding PCC delivery**

*Reception.*  Almost all participants expressed a positive emotional response toward receiving clients seeking preconception care (PCC) services. They reported feeling pleased and fulfilled when individuals presented for such care. However, they also noted that it was relatively uncommon for clients to proactively seek PCC. Despite its rarity, participants consistently conveyed their willingness and enthusiasm to welcome these clients and deliver the necessary services to meet their needs and ensure their satisfaction. Here is what the participants said:

*"Yeah, I think they are even the people we welcome more because it's scarce to get such people coming closer."* [A Senior Midwifery Officer at the ANC, P1]

*"Usually, we are always happy to receive the person because not everybody that has the time to come for preconception care before becoming pregnant, so anytime we receive somebody coming for preconception services, we are always happy and we provide all the information necessary to help the person."* [A Senior Staff Midwife at the ANC, P2]

*"Oh! We're always happy to receive them, especially when you're able to take the person through and see the person happily pregnant and deliver, you even become happy…"* [A Senior Staff Midwife at the Maternity, P10]

**Midwives' willingness to provide PCC.**  The majority of participants affirmed that the provision of PCC services falls within the scope of their professional responsibilities as midwives. Consequently, they expressed consistent availability and a strong willingness to deliver PCC to their clients whenever needed. Here are some statements made by the participants:

*"…is part of the midwife's role at the ANC to provide preconception care to their clients."* [A Senior Staff Midwife at the ANC, P2]

*"Is our responsibility to provide regardless of your mood…even if you don't want to, you have to provide because that's part of our job description…."* [A Staff Midwife at the ANC, P6]

Conversely, some participants acknowledged that, despite recognizing PCC as part of their professional responsibilities, they frequently face challenges in delivering these services due to limited staffing and demanding workloads. As a result, they often prioritize critical or emergency cases over clients seeking PCC, which may lead to inconsistent provision of preconception care. The following participants expressed this:

*"As a midwife, I think that is the problem of the day because a lot of people do not get time to deliver these services [PCC], but it's necessary and important for us to, because we need to get time, we need to sacrifice…"* [A Senior Staff Midwife at the ANC, P2]

*"…. currently, where I am, am the only midwife, so let's say on a day that the community health nurses are having their CWC or going out for outreach and there is pressure on me…. attending to those clients is always difficult because I would be alone."* [A Staff Midwife at a Health Center, P9]

*"So, sometimes you are busy doing other activities, so what we do is that you schedule the person to your free time, so you'll have time for them. Sometimes, if you're the only midwife at the facility and there are pregnant women there, you won't get time for the one coming for preconception care."* [A Senior Staff Midwife at a Health Center, P14]

**Communication.** Most participants reported that, given the infrequent utilization of PCC services, they prioritize making the most of each opportunity to provide comprehensive education and counseling. They emphasized delivering care with respect and dignity, engaging clients in calm and reassuring communication to build trust. This approach facilitates open dialogue, encouraging clients to share relevant information and concerns freely. Here are statements from some of the participants:

*"…when talking to them, we look in their faces and then speak in a calm manner…. we bring words that they will understand, and then you should also be willing to help, not to always shout or be harsh on them during the counseling. Because your attitude plays a role, it helps the woman to open up to tell you a lot."* [A Midwifery Officer at the Maternity, P3]

*"…is like one-to-one interaction, is not like you would just have to pump everything into them, no. Is a counseling session, and you don't have to impose certain things on them; they will bring their opinion then you take it up."* [A Staff Midwife at a Health Center, P9]

*"… you shouting would just aggravate everything, so we talk to them nicely so that they would also have some trust that things would work well."* [A Senior Staff Midwife at a Health Center, P14]

**PCC delivery practices among midwives**

***Provision of PCC.*** Most participants indicated that they do provide PCC services; however, these services are not offered regularly, as clients typically seek PCC only when experiencing difficulties with conception. The participants reported that their PCC practices primarily involve counseling, conducting laboratory investigations, and referring clients to specialist care when necessary. The following participants shared their experiences:

*"…. those that I've met, most of them are in need of children, so what I do is to refer them to see a doctor, but before that, I send them to the laboratory for screening and other things. Some of them have their hormones may be too high or too low, thus either this estrogen or progesterone, and you know it [hormones] can affect childbearing. So, if it is too low, definitely, they have to get help, and if it is high, then you've to work on it."* [A Senior Staff Midwife at the Maternity, P4]

*"There was this woman who got married and was not conceiving for some time, so it became a border and a worry to her. So, she came to me and then spoke about it. From the conversation, I got to know that her menses wasn't regular, so I referred her to see a medical doctor at the district hospital…."* [A Staff Midwife at a Health Center, P7]

*"There was this couple that came to me when I was posted, and the issue was that anytime the lady picks seed, she gets a miscarriage, so they came for help. I took them through counseling and then made them go for lab*

*investigations. When the results came, it was due to rhesus incompatibility, so I referred them to see a specialist in Tamale. Thanks be to God, now they have a baby boy who is getting to 1 year now."* [A Staff Midwife at a Health Center, P9]

**A dedicated unit for PCC delivery.** Most participants reported the absence of a designated area or dedicated unit specifically for PCC services. They indicated that PCC is typically integrated and delivered alongside other services, particularly within antenatal care (ANC) or reproductive and child health (RCH) units. The following participants said:

*"For now, we don't have a designated unit for the preconception care…. we combine and do it together with the other clients."* [A Senior Staff Midwife at the ANC, P2]

*"Specifically, we don't, we don't have a unit that probably we've labeled it that for preconception care, so we add that service to our antenatal care. So, anyone who needs that service comes to us at the antenatal for it."* [A Staff Midwife at the ANC, P6]

*"Ideally, you are supposed to have a room so that we can provide the services there, but in our setting, we don't have those facilities."* [A Senior Staff Midwife at a Health Center, P14]

## Discussion

The study participants demonstrated good knowledge of PCC, encompassing its definition, components, benefits, and intended beneficiaries. These findings are consistent with Ukoha & Mtshali [39], where 88% of primary healthcare nurses exhibited good knowledge of PCC. The range of services identified under PCC in this study included screening, health promotion, risk and nutritional assessments, counseling, supplementation, and the management of medical and obstetrical conditions. Similarly, healthcare providers in previous studies recognized key PCC components such as genetic screening, counseling, family planning services, and the identification and management of health conditions or potential risks [40,41]. Consistent with the present study, the literature identifies the primary beneficiaries of PCC as couples, adolescents, individuals with comorbidities, and those planning pregnancies [1,42,43]. Particular emphasis was placed on adolescents and individuals with chronic conditions, such as HIV, sickle cell disease (SCD), hypertension, and diabetes, who require tailored PCC interventions [44]. Participants also demonstrated an understanding of the benefits of PCC, including facilitating prospective parents' awareness of their health and blood status prior to conception. These findings corroborate previous studies that emphasize the positive outcomes associated with PCC services [21,22,45,46].

The study revealed that midwives generally exhibited positive attitudes toward the delivery of preconception care (PCC), with nearly all participants expressing satisfaction and enthusiasm when receiving clients seeking PCC services at their units. This favorable reception contributed to creating a welcoming and supportive environment for clients. These findings are consistent with those of Abayneh and colleagues [21], who reported similarly positive attitudes toward PCC among obstetric healthcare providers. Existing literature suggests that positive healthcare provider attitudes significantly enhance the utilization of PCC services, whereas negative attitudes constitute substantial barriers to uptake [47]. The positive attitudes identified among study participants encompassed professional support, reassurance, encouragement, respect, trust, and confidentiality. Conversely, negative attitudes that were perceived to hinder PCC utilization included judgmental behavior, unreliability, stereotyping, restrictiveness, lack of support, authoritarianism, and poor communication skills.

Some participants in this study reported providing preconception care (PCC) services; however, these were delivered irregularly due to low client demand. This finding aligns with a systematic review by Poels and colleagues [47], which identified that most women were not offered PCC because information about such care was not routinely integrated into daily healthcare practices. Similarly, Kassa and colleagues [26] found that a significant majority (85%) of healthcare providers

(HCPs) did not practice PCC regularly, with only 15% offering it sporadically or not at all. Notably, nurses and midwives were almost twice as likely as medical doctors to refrain from practicing PCC. Consistent with these findings, Abayneh and colleagues [21] also reported suboptimal PCC practices among HCPs. In the present study, participants indicated that PCC provision was often contingent upon the specific needs of clients rather than delivered as a routine service. This reflects a broader trend documented in the literature, where PCC implementation among nurses and other healthcare professionals remains low [6,15,45]. Despite most participants engaging in PCC activities, regular practice was hindered by challenges such as the absence of clearly defined protocols, guidelines, and dedicated service units for PCC delivery. These barriers are echoed in other studies highlighting that the lack of standardized guidelines creates uncertainty among HCPs regarding their roles and responsibilities in PCC provision [3,18,48]. Kassa and colleagues [26] further emphasized that unclear responsibilities were a significant factor in the non-implementation of PCC services. Some researchers have proposed that family planning clinics could function as effective platforms for PCC delivery in the absence of dedicated units [39,49,50]. To enhance PCC uptake and quality, it is imperative to develop and disseminate standardized guidelines and protocols that clearly define healthcare providers' roles, alongside establishing dedicated PCC units staffed by trained personnel.

## Conclusion

This study underscores the substantial knowledge and good attitudes midwives possess regarding preconception care, including its components, benefits, and target groups. Despite this strong foundation, the delivery of PCC services remains inconsistent, largely due to the lack of standardized protocols and dedicated units for PCC. This gap in structure leads healthcare providers to rely on personal knowledge and ad hoc practices, resulting in uneven implementation. The study highlights the need for clear guidelines and protocols to standardize PCC delivery, ensuring that all individuals within reproductive age groups, particularly those with specific health conditions, receive preconception care before getting pregnant. Additionally, fostering good attitudes among healthcare providers is crucial, as these attitudes significantly influence the utilization of PCC services. Addressing these challenges through policy development and training can enhance the consistency and effectiveness of PCC, ultimately improving maternal and child health outcomes.

## Supporting information

**S1 Data.  Full transcription of qualitative interviews with participants, detailing their responses regarding the availability of preconception care services in health facilities.**
(DOCX)

## Acknowledgments

We would like to thank all participants who volunteered in our study. We also sincerely thank the Kpandai District Health Director and her staff for allowing us to conduct this study.

## Author contributions

**Conceptualization:** Richard Kwaku Bawah, Barnabas Sackeyuah.

**Data curation:** Richard Kwaku Bawah, Barnabas Sackeyuah.

**Formal analysis:** Richard Kwaku Bawah, Barnabas Sackeyuah.

**Investigation:** Richard Kwaku Bawah, Barnabas Sackeyuah.

**Methodology:** Richard Kwaku Bawah, Barnabas Sackeyuah.

**Supervision:** Gilbert Abotisem Abiiro, Gifty Apiung Aninanya.

**Validation:** Gilbert Abotisem Abiiro, Gifty Apiung Aninanya.

**Writing – original draft:** Richard Kwaku Bawah, Barnabas Sackeyuah.

**Writing – review & editing:** Gilbert Abotisem Abiiro, Gifty Apiung Aninanya.

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
