## [Decision Letter · Decision Letter 0]

22 Jan 2025

PGPH-D-24-02270

An exploratory study on midwives' perceptions of the availability of preconception care in health facilities in the Kpandai District of Northern Ghana.

Dear Dr. Bawah,

Thank you for submitting your manuscript to PLOS Global Public Health. After careful consideration, we feel that it has merit but does not fully meet PLOS Global Public Health’s publication criteria as it currently stands. Therefore, we invite you to submit a revised version of the manuscript that addresses the points raised during the review process.

**Editorial Comments:**

**Justification for Sampling Methods:** Please provide a clear and detailed justification for the use of both purposive and random sampling techniques within your qualitative study. Explain how these methods align with your study's objectives and contribute to the depth and breadth of the data collected. Additionally, discuss the rationale for integrating these sampling approaches and how they complement each other in achieving a comprehensive understanding of the research phenomena.**Description of Manual Thematic Coding:** The description of your manual thematic coding process requires further elaboration. Provide a step-by-step explanation of how you approached coding the data manually. For instance: Detail how you identified themes or patterns, specify any frameworks, criteria, or guidelines used during the coding process and describe how codes were reviewed, refined, and organized into broader themes.

To strengthen this section, consider referencing articles or methodologies that offer established and detailed processes for manual coding in qualitative research. For example: https://doi.org/10.1177/16094069241299 . This will enhance the clarity and credibility of your methodology.

**Reviewer comments**

I feel that this paper does provide valuable insight into the PCC, which is very important in addressing the global low fertility rate. The authors can be consistent in using terms such as "good attitude" or "positive" vs. favorable.

We look forward to receiving your revised manuscript.

Kind regards,

Kahabi Ganka Isangula, MD, MPH, PhD

Academic Editor

Journal Requirements:

2. Please provide an Author Summary. This should appear in your manuscript between the Abstract (if applicable) and the Introduction, and should be 150–200 words long. The aim should be to make your findings accessible to a wide audience that includes both scientists and non-scientists. Sample summaries can be found on our website under Submission Guidelines:

https://journals.plos.org/globalpublichealth/s/submission-guidelines#loc-parts-of-a-submission.

3. In the online submission form, you indicated that "As this study is qualitative, the data generated are not publicly accessible. However, additional information regarding the data can be obtained from the corresponding author upon reasonable request.". 

a. In a public repository, 

b. Within the manuscript itself, or 

c. Uploaded as supplementary information.

Additional Editor Comments (if provided):

Decision:

We are pleased to accept your manuscript pending minor revisions.

Editorial Comments:

1. Justification for Sampling Methods: Please provide a clear and detailed justification for the use of both purposive and random sampling techniques within your qualitative study. Explain how these methods align with your study's objectives and contribute to the depth and breadth of the data collected. Additionally, discuss the rationale for integrating these sampling approaches and how they complement each other in achieving a comprehensive understanding of the research phenomena.

2. Description of Manual Thematic Coding: The description of your manual thematic coding process requires further elaboration. Provide a step-by-step explanation of how you approached coding the data manually. For instance: Detail how you identified themes or patterns, specify any frameworks, criteria, or guidelines used during the coding process and describe how codes were reviewed, refined, and organized into broader themes.

To strengthen this section, consider referencing articles or methodologies that offer established and detailed processes for manual coding in qualitative research. For example: https://doi.org/10.1177/16094069241299 . This will enhance the clarity and credibility of your methodology.

Reviewer comments

I feel that this paper does provide valuable insight into the PCC, which is very important in addressing the global low fertility rate. The authors can be consistent in using terms such as "good attitude" or "positive" vs. favorable.

Reviewers' comments:

Reviewer's Responses to Questions

**Comments to the Author**

1. Does this manuscript meet PLOS Global Public Health’s publication criteria ? Is the manuscript technically sound, and do the data support the conclusions? The manuscript must describe methodologically and ethically rigorous research with conclusions that are appropriately drawn based on the data presented.

Reviewer #1: Yes

2. Has the statistical analysis been performed appropriately and rigorously?

Reviewer #1: N/A

3. Have the authors made all data underlying the findings in their manuscript fully available (please refer to the Data Availability Statement at the start of the manuscript PDF file)?

Reviewer #1: Yes

4. Is the manuscript presented in an intelligible fashion and written in standard English?

Reviewer #1: Yes

5. Review Comments to the Author

Reviewer #1: I feel that this paper does provide valuable insight into the PCC, which is very important in addressing the global low fertility rate. The authors can be consistent in using terms such as "good attitude" or "positive" vs. favorable.

6. PLOS authors have the option to publish the peer review history of their article (what does this mean? ). If published, this will include your full peer review and any attached files.

**Do you want your identity to be public for this peer review?** For information about this choice, including consent withdrawal, please see our Privacy Policy .

Reviewer #1: No

---

## [Decision Letter · Decision Letter 1]

20 May 2025

PGPH-D-24-02270R1

An exploratory study on midwives' perceptions of the availability of preconception care in health facilities in the Kpandai District of Northern Ghana.

Dear Dr. Bawah,

Thank you for submitting your manuscript to PLOS Global Public Health. After careful consideration, we feel that it has merit but does not fully meet PLOS Global Public Health’s publication criteria as it currently stands. Therefore, we invite you to submit a revised version of the manuscript that addresses the points raised during the review process.

We look forward to receiving your revised manuscript.

Kind regards,

Paolo Angelo Cortesi, PhD

Academic Editor

Journal Requirements:

Additional Editor Comments (if provided):

Reviewers' comments:

Reviewer's Responses to Questions

**Comments to the Author**

1. If the authors have adequately addressed your comments raised in a previous round of review and you feel that this manuscript is now acceptable for publication, you may indicate that here to bypass the “Comments to the Author” section, enter your conflict of interest statement in the “Confidential to Editor” section, and submit your "Accept" recommendation.

Reviewer #2: (No Response)

2. Does this manuscript meet PLOS Global Public Health’s publication criteria ? Is the manuscript technically sound, and do the data support the conclusions? The manuscript must describe methodologically and ethically rigorous research with conclusions that are appropriately drawn based on the data presented.

Reviewer #2: Partly

3. Has the statistical analysis been performed appropriately and rigorously?

Reviewer #2: Yes

4. Have the authors made all data underlying the findings in their manuscript fully available (please refer to the Data Availability Statement at the start of the manuscript PDF file)?

Reviewer #2: Yes

5. Is the manuscript presented in an intelligible fashion and written in standard English?

Reviewer #2: No

6. Review Comments to the Author

Reviewer #2: 1. Please simply the language suitable for a science journal, not a newspaper style. Example "....before they embark on the journey of pregnancy."

2. "Midwives perceptions of the availability of preconception care" appears to be distal indicator. Ideally if it has any value for programs, it should start with an inclusive review of all aspects of preconception care availability not just midwives "perception". In addition to the midwives, there are nurses, and other clinical service providers, ministry of health guidance, fidelity of the integration of such guidance, and health care seeking behaviors of women of childbearing age. If health care seeking prior to conception is not a practice, how would midwives intervene? Do the midwives go and talk to each women of childbearing age? Or do you have a preconception counseling at the time of marriage registration as it was in China, leading to the studies on folic acid supplementation? So, what good is to study the perceptions of a fraction of service providers, when we do not know if all other elements of preconception care are in place?

7. PLOS authors have the option to publish the peer review history of their article (what does this mean? ). If published, this will include your full peer review and any attached files.

**Do you want your identity to be public for this peer review?** For information about this choice, including consent withdrawal, please see our Privacy Policy .

Reviewer #2: No

---

## [Editor Report · Decision Letter 2]

22 Jul 2025

An exploratory study on midwives' perceptions of the availability of preconception care in health facilities in the Kpandai District of Northern Ghana.

PGPH-D-24-02270R2

Dear Mr Bawah,

We are pleased to inform you that your manuscript 'An exploratory study on midwives' perceptions of the availability of preconception care in health facilities in the Kpandai District of Northern Ghana.' has been provisionally accepted for publication in PLOS Global Public Health.

Best regards,

Paolo Angelo Cortesi, PhD

Academic Editor